# Presenilin-2 and Calcium Handling: Molecules, Organelles, Cells and Brain Networks

**DOI:** 10.3390/cells9102166

**Published:** 2020-09-25

**Authors:** Paola Pizzo, Emy Basso, Riccardo Filadi, Elisa Greotti, Alessandro Leparulo, Diana Pendin, Nelly Redolfi, Michela Rossini, Nicola Vajente, Tullio Pozzan, Cristina Fasolato

**Affiliations:** 1Department of Biomedical Sciences, University of Padua, Via U. Bassi 58/B, 35131 Padua, Italy; emy.basso@cnr.it (E.B.); riccardo.filadi@unipd.it (R.F.); elisa.greotti@cnr.it (E.G.); alessandro.leparulo@unipd.it (A.L.); diana.pendin@unipd.it (D.P.); nelly.redolfi@unipd.it (N.R.); michela.rossini@studenti.unipd.it (M.R.); nicola.vajente@studenti.unipd.it (N.V.); tullio.pozzan@unipd.it (T.P.); 2Neuroscience Institute, Italian National Research Council (CNR), Via U. Bassi 58/B, 35131 Padua, Italy; 3Venetian Institute of Molecular Medicine (VIMM), Via G. Orus 2B, 35131 Padua, Italy

**Keywords:** presenilin-2, calcium signalling, Alzheimer’s disease mouse models, SOCE, mitochondria, autophagy, brain networks, oscillations, slow-waves, functional connectivity

## Abstract

Presenilin-2 (PS2) is one of the three proteins that are dominantly mutated in familial Alzheimer’s disease (FAD). It forms the catalytic core of the γ-secretase complex—a function shared with its homolog presenilin-1 (PS1)—the enzyme ultimately responsible of amyloid-β (Aβ) formation. Besides its enzymatic activity, PS2 is a multifunctional protein, being specifically involved, independently of γ-secretase activity, in the modulation of several cellular processes, such as Ca^2+^ signalling, mitochondrial function, inter-organelle communication, and autophagy. As for the former, evidence has accumulated that supports the involvement of PS2 at different levels, ranging from organelle Ca^2+^ handling to Ca^2+^ entry through plasma membrane channels. Thus FAD-linked PS2 mutations impact on multiple aspects of cell and tissue physiology, including bioenergetics and brain network excitability. In this contribution, we summarize the main findings on PS2, primarily as a modulator of Ca^2+^ homeostasis, with particular emphasis on the role of its mutations in the pathogenesis of FAD. Identification of cell pathways and molecules that are specifically targeted by PS2 mutants, as well as of common targets shared with PS1 mutants, will be fundamental to disentangle the complexity of memory loss and brain degeneration that occurs in Alzheimer’s disease (AD).

## 1. Presenilin-2 in Physiology and Pathology

Presenilin-2 (PS2)—and its homolog presenilin-1 (PS1)—is a 50-kDa multi-pass membrane protein with nine helical transmembrane (TM) domains, and in humans it is encoded by a gene present on chromosome 1 (*PSEN2*) [1]. Both presenilins (PSs) mainly localize to the endoplasmic reticulum (ER) and Golgi apparatus (GA) membranes but also, although less abundantly, in plasma membrane (PM) and endosomes [2]. Their mRNAs are expressed in different human and mouse tissues, with the highest levels in the hippocampus and cerebellum [3].

Both PSs represent the catalytic core of the γ-secretase complex, the enzyme ultimately responsible for generation of Aβ peptides; they were both discovered in genetic analyses of families in which Alzheimer’s disease (AD) is transmitted as an autosomal dominant trait. In fact, as of now, about 300 mutations in *PSEN1* and 58 mutations in *PSEN2* have been described (https://www.alzforum.org/mutations), the majority of which are dominant, mostly missense, and have been associated with the inherited forms of the disease (familial Alzheimer’s disease (FAD)) [4,5]. Mutations in the gene for one of the substrate of the γ-secretase complex, the amyloid precursor protein (APP), are also responsible for FAD cases [6]. It has been proposed that FAD-PS mutations lead to a less precise γ-secretase cleavage of APP, in some cases decreasing the total production of Aβ but increasing the relative amount of the more amyloidogenic Aβ42 peptide, the seeding core of extracellular amyloid plaques, over the more soluble Aβ40 peptide [7,8].

The γ-secretase complex is part of the family of intramembrane-cleaving proteases (I-CliPs), which perform hydrolysis of protein domains embedded in the hydrophobic environment of the membrane. The family includes SP2 metalloproteases, serine proteases of the rhomboid family, and the aspartyl proteases to which γ-secretase belongs.

The γ-secretase has a central role in cellular biology, with about 150 different integral membrane proteins recognized as substrates [9]; the most studied are the Notch family of receptors, with a crucial role in signalling and cell differentiation, and APP [4,9]. The γ-secretase complex is composed of four subunits: PS1 or PS2; nicastrin, an integral membrane protein concerned with substrate recognition and selection [10]; PS enhancer-2 (PEN-2) that stabilizes the PS complex and has a role in its endoproteolytic cleavage [11,12,13]; and anterior pharynx defective 1 (APH1), which interacts with nicastrin, providing the initial scaffold to which PS1/2 and PEN-2 are added [14,15]. In humans, APH1 is encoded by two paralogous genes (APH1A and APH1B), and each protein can interact with either PS, resulting in the existence of four different γ-secretase complexes that might have slightly different specificities [16]. After its enclosure within the complex, PS undergoes an endoproteolytic cleavage that produces N- and C-terminal fragments; the two fragments remain associated and represent the biologically active form of the complex, each carrying one of the two key aspartic acid residues on TM6 and TM7, respectively [17,18].

PS1 and PS2 share about 66% of amino acidic sequence; one key difference is a motif in PS2 that interacts with activating protein-1 (AP-1) complexes in a phosphorylation-dependent manner and targets PS2 to the late endosome/lysosome compartment, leading to a different subcellular distribution of PS2 and perhaps to subtly different functions [19,20]. For example, it has been demonstrated that PS2-containing γ-secretase complexes are involved in the processing of premelanosome (PMEL) protein, which is involved in melanosome maturation and melanin deposition [19]. Indeed, PS2-null zebrafish showed defects in skin pigmentation [21]. Importantly, melanosome biogenesis seems to be Ca^2+^-dependent [22] (see also below).

Several γ-secretase-independent functions of PSs have emerged in the recent years, enriching the overall importance of these proteins in cell biology. For example, PSs bind to glycogen synthase kinase 3β (GSK3β), a key protein of the Wnt signalling pathway, and to its substrate β-catenin, a transcription regulator [23,24]. The interaction of PSs with GSK3β and β-catenin is independent of γ-secretase activity [25] and influences β-catenin phosphorylation and turnover [26], as well as the activity of kinesin-1 and dynein and thus axonal transport of type 1 transmembrane receptors [27]. PSs have been implicated also in autophagy (see below) and protein trafficking [28].

Last, but not least, the regulation of cellular Ca^2+^ homeostasis has emerged as a key PS function, independent of γ-secretase activity, with relevant implications in multiple Ca^2+^-regulated cell processes. In the present review, we summarize the central role played by PS2 in cellular Ca^2+^ homeostasis, highlighting divergent and convergent aspects of PS2 vs. PS1 pathophysiology.

## 2. PS2 and Ca^2+^ Homeostasis

### 2.1. Alterations of Ca^2+^ Homeostasis in FAD-PS2 Cell Models

According to the so-called “Ca^2+^ overload” hypothesis for AD, FAD-PS mutations increase the ER Ca^2+^ content and cause excessive cytosolic Ca^2+^ release upon cell stimulations that, in turn, alters APP processing and sensitizes neurons to Ca^2+^-dependent cell death mechanisms [29]. Indeed, an excessive release of Ca^2+^ from the ER has been reported in different cell models expressing various FAD-PS mutations, as well as in neurons from transgenic (Tg) mice carrying FAD-PS1 mutations [30,31,32,33,34,35]. FAD-linked mutations in PS2 have also been reported to potentiate ER Ca^2+^ release from both ryanodine receptors (RyRs) [36] and inositol trisphosphate (IP3) receptors (IP3Rs) in *Xenopus* oocytes [30] and neurons from Tg mice expressing the PS2-N141I mutation [37]. Moreover, it has been proposed that wild type (WT) PSs form constitutively active ER Ca^2+^ leak channels whereas FAD-PS mutations disrupt the channel functionality; as a result of the reduced leak, the ER Ca^2+^ level increases and more Ca^2+^ is released upon stimulation [38].

In contrast, we showed that FAD patient-derived fibroblasts carrying the PS2-M239I mutation, as well as HeLa and HEK293 cells stably or transiently expressing the same PS2 mutant, show a decreased ER Ca^2+^ release when stimulated by IP3-generating agonists [39] (Figure 1); this result was confirmed in FAD patient-derived fibroblasts carrying another PS2 mutation (T122R) [40]. Of note, in this study, we analysed two monozygotic twins, one with overt signs of disease at the time of biopsy, whereas the other one was still asymptomatic; nevertheless, both cell samples shared a similar Ca^2+^ handling defect, strongly suggesting that Ca^2+^ dysregulation represents an early event in the pathogenesis of AD [40].

To clarify the divergent results, we directly monitored Ca^2+^ dynamics within intracellular stores. We employed aequorin-based Ca^2+^ probes targeted to ER and GA in cells expressing different PS1 and PS2 mutants. In several cell lines [SH-SY5Y, HeLa, HEK293 and Mouse Embryonic Fibroblast (MEF) cells], expressing the ER (or GA)-targeted aequorin together with a number of PS1 (P117L, M146L, L286V, and A246E) or PS2 (M239I, T122R, and N141I) mutants, we analysed Ca^2+^ concentrations and dynamics in the two organelles. By this more specific approach, we confirmed lower ER and GA Ca^2+^ levels in the presence of all the analysed FAD-PS2 mutants, and unchanged or slightly decreased ER/GA Ca^2+^ concentrations when PS1 mutants were expressed [41]. Similar results were obtained in FAD patient-derived fibroblasts and rat primary neurons, expressing either PS1 or PS2 mutants and loaded with the Ca^2+^ sensor fura-2, confirming the capability of PS to modify Ca^2+^ homeostasis, but questioning the “Ca^2+^ overload” hypothesis for AD [41,42]. Indeed, it was also shown that FAD-PS are associated with IP3R hyperactivity [43,44], providing an alternative explanation to the “Ca^2+^ overload” hypothesis based on increased ER Ca^2+^ release findings previously reported in FAD-PS-expressing cells. In particular, Foskett and co-workers showed that FAD-PS1/2 mutants, by physically interacting with the IP3R, modulate the channel gating, causing an exaggerated ER Ca^2+^ release regardless of its Ca^2+^ content [43,44]. Furthermore, by employing ER- and GA-targeted Ca^2+^ indicators, the same group subsequently confirmed that cells expressing FAD-PS1 mutants do not present ER Ca^2+^ overload, arguing against the previously proposed role of PS as ER Ca^2+^ leak channels [45].

Similarly, by using newly developed genetically encoded Ca^2+^ indicators, the Förster resonance energy transfer (FRET)-based probe targeted to ER (D4ER [46]), medial-GA [47], and trans-GA [48], we showed that (i) in SH-SY5Y and Baby Hamster Kidney (BHK) cells, expressing the FAD-PS2-T122R mutant, and in PS2-N141I patient-derived fibroblasts, there is a clear reduction in ER Ca^2+^ concentration; (ii) in cells expressing the FAD-PS1-A246E mutant, instead, no change was observed [46,49]; (iii) the expression of FAD-PS2 mutants induced a selective decrease in the medial-GA Ca^2+^ content, but not in that of the trans-GA; (iv) in contrast, the expression of the FAD-PS1 mutant was ineffective on both GA sub-compartments [49] (Figure 1). 

Concerning the molecular mechanism through which FAD-PS2 alters intracellular Ca^2+^ store dynamics, researchers have shown that the Ca^2+^ phenotype is caused by the holoprotein and that it is independent of γ-secretase activity [38,39,42,44,49,50,51]. Moreover, it has been shown that the protein directly interacts with the IP3R, sensitizing it to lower IP3 concentrations [43], the RyR [36], the RyR-regulating protein sorcin [52], and the SERCA2b [34], inhibiting its activity [51]. This latter result is consistent with the differential effect of FAD-PS2 mutants on GA sub-compartments (see above), given that the trans-GA, where FAD-PS2 mutants are ineffective, relies only on the secretory pathway Ca^2+^ ATPase 1 (SPCA1) for Ca^2+^ uptake [48]. Finally, in the presence of FAD-PS1/2 mutations, increased expression levels and activity of RyRs have been reported [53,54,55], suggesting a RyR-dependent Ca^2+^ hyperexcitability in AD that is antagonized by the channel inhibitor dantrolene (see [56] for an extensive discussion of this issue; see also [57] for the involvement of IP3Rs). 

Intracellular Ca^2+^ stores, mainly the ER, are functionally and physically coupled to mitochondria with which they jointly operate modulating several cell functionalities, such as lipid synthesis and Ca^2+^ homeostasis. Specific ER membrane domains tightly juxtaposed to mitochondria, called mitochondria-associated membranes (MAM; [58]), represent signalling platforms and play a key role in these processes [59]. Interestingly, MAM appear to be altered in AD samples [42,54,59,60,61,62,63,64]; in addition PS1/2, as well as the other components of the γ-secretase complex and APP, are enriched in these domains [63,65,66]. Only FAD-PS2 mutants, however, are able to increase the interaction between the two organelles, facilitating ER–mitochondria Ca^2+^ transfer [42,54,63] by binding to mitofusin-2 [63] and thus removing its negative modulation on organelle tethering [67] (Figure 1).

The other Ca^2+^ signalling pathway affected by FAD-PS2 is the store-operated Ca^2+^ entry (SOCE) [68,69]. In particular, it has been shown that several FAD-PS2 mutants reduce SOCE activity in different cell types [40,41,49,70]. Interestingly, this effect is shared with FAD-PS1 mutants, which similarly reduce this Ca^2+^ influx [41,49,70,71] (Figure 1). Accordingly, SOCE is potentiated in cells where PS levels are reduced [70,72]. In PS double knock out (KO) MEFs and in B-lymphocytes derived from patients expressing FAD-PS mutants [73], researchers have found that the levels of the key SOCE components Stromal interaction molecule (STIM) STIM1 and STIM2 [68,69] are reduced. Of note, alterations in SOCE and STIM1 protein level have also been reported in sporadic AD (SAD) patients [74]. It has been proposed that SOCE is regulated by a γ-secretase-dependent mechanism, with STIM1 being a substrate of PS1-containing γ-secretase complexes [71]. Nevertheless, we found lower SOCE and STIM1 protein levels in both FAD-PS1- and FAD-PS2-expressing cells treated with the γ-secretase inhibitor DAPT [49] (Figure 1). 

Of note, the overexpression of WT-PS2 often mimics the effect of its FAD mutants on Ca^2+^ homeostasis, although higher levels of WT-PS2 are required to obtain the alterations in Ca^2+^ homeostasis elicited by FAD-PS2 mutants [40]. This latter finding could be relevant for SAD forms of the disease, where an upregulation of the endogenous PS2 has been reported in brain AD samples due to the loss of repressor element 1-silencing transcription factor (REST) [75]. It can be speculated that an abnormal accumulation of PS2 holoprotein could cause Ca^2+^ signalling dysregulation, also typically observed in SAD cases.

### 2.2. Calcium Handling in AD Mouse Models Expressing PS2-N141I

The findings reported above led us to investigate Ca^2+^ handling and brain network functionality in Tg mouse lines based on FAD-PS2 mutants by means of in vitro and in vivo approaches. We took advantage of two homozygous mouse lines expressing the PS2-N141I mutant, as described in detail in Box 1: the double Tg line B6.152H, also known as B6.PS2APP, and the single Tg line PS2.30H [76]. Here, we simply refer to these two lines as 2TG and TG, respectively.

We were firstly interested to verify whether the same Ca^2+^ changes found in FAD-PS2-expressing cell lines were also detectable in primary neuronal cultures and in acute hippocampal slices from 2-week-old animals. At this age, total brain Aβ levels in 2TG mice are still very low, but are already detectable and higher when compared to WT and TG mice [54]. Both TG and 2TG neurons, in culture or in situ, show a reduction in the ER Ca^2+^ content, when estimated indirectly, through Ca^2+^ release induced by IP3-generating agonists, or directly, with the Cameleon probe D4ER [46,54]. 

In acute hippocampal slices, upon stimulation with IP3-generating agonists, Ca^2+^ release was dramatically reduced not only in neurons but also in astrocytes of TG and 2TG mice, suggesting defective store Ca^2+^ content, as well as Ca^2+^ entry, in these latter cell types [54]. Importantly, these changes occur precociously and independently of APP overexpression and brain Aβ load, being found equally in TG and 2TG mice; thus, they reflect the intrinsic capability of modulating Ca^2+^ handling of FAD-PS2 mutants. 

Studying neurons in vitro and in situ allowed us to also highlight relevant network properties brought about by the PS2 mutant. In both conditions, neuronal cells, when exposed to picrotoxin, a γ-aminobutyric acid (GABA)-A receptor antagonist, showed synchronous Ca^2+^ spiking activity that was higher in TG and 2TG mice with respect to WT [54]. This type of Ca^2+^ spiking is independent of Ca^2+^ stores and likely due to an imbalance between excitatory and inhibitory inputs that represent an early sign of network dysfunction [77,78]. 

## 3. Functional Effects of Ca^2+^ Dysregulation by FAD-PS2

### 3.1. Autophagy

Macroautophagy (hereafter autophagy) is a process in which double-membrane vesicles (called autophagosomes) engulf different cellular components (including misfolded proteins, portions of cytosol, and damaged organelles) and target them to lysosomes, where they are degraded into simpler molecular constituents. 

In 2004, two seminal papers firstly suggested that PSs might be involved in autophagy modulation [79,80]. Specifically, Esselens and co-workers reported telencephalin accumulation within autophagosomes in PS1-KO hippocampal neurons as a result of a defective fusion of these vesicles with lysosomes. Similarly, Wilson and colleagues observed that PS1 deficiency, in fibroblasts and primary cortical neurons, resulted in the formation of enlarged lysosomes, with accumulation of α- and β-synuclein. Importantly, this phenomenon was likely associated with SOCE augmentation, suggesting that altered Ca^2+^ signalling may underpin the effect of PSs on the autophagy pathway [79]. Additional investigations, mostly focused on PS1, consistently reported that PSs modulate autophagosome-lysosome fusion. Nevertheless, consensus has not been reached on the underlying mechanism. Indeed, either a defective lysosomal acidification [81], a reduced lysosomal Ca^2+^ content [82], or altered expression of key genes belonging to the coordinated lysosomal expression and regulation (CLEAR) network [83,84] have been suggested as possible mechanisms (reviewed in [85]). 

As far as FAD-PS mutants are concerned, some of the discrepancies might be linked to mutation-specific effects. Nevertheless, most studies converge on the lack of involvement of the γ-secretase activity, whereas Ca^2+^ signalling dysregulation has been frequently reported as a common feature among different FAD-PS mutants [85]. Recently, we observed that the reduced ER Ca^2+^ content, consistently observed in different FAD-PS2 cell models, affects the fusion of autophagosomes with lysosomes, thus inducing autophagosome accumulation [86] (Figure 2). Specifically, the phenomenon appears linked to the generation of lower cytosolic Ca^2+^ rises upon IP3-induced release of ER Ca^2+^, given that it can be mimicked by increasing the cytosolic Ca^2+^-buffering capacity (loading cells with the permeable forms of Ca^2+^ chelating agents). Mechanistically, we found that alterations of cytosolic Ca^2+^ dynamics affect the recruitment to autophagosomes of Ras-associated binding protein RAB7, a small GTPase whose association with both autophagosomes and lysosomes tunes their fusion in the final steps of the autophagy pathway [87]. Importantly, at variance with previous studies focused on FAD-PS1 [81,82,88], neither the pH of lysosomes nor their Ca^2+^ content were found to be affected by FAD-PS2 mutants [86]. Taken together these observations suggest that slightly different mechanisms might underlie the effects of FAD-PS1 and FAD-PS2 on the autophagy flux, with an altered Ca^2+^ signalling (though by distinct pathways) being a common feature. 

### 3.2. Cell Metabolism and Bioenergetics

The first piece of evidence that WT-PS2 modulates mitochondrial metabolism was found in 2006, when lower mitochondrial respiration and decreased mitochondrial membrane potential (Δψm) were observed in *PSEN2^−/−^*, but not in *PSEN1^−/−^* MEFs [89]. Later, similar results were obtained by Contino and co-investigators [90], who found reduced basal and maximal mitochondrial oxygen consumption in *PSEN2^−/−^* and PS double KO MEFs (but not in *PSEN1^−/−^* MEFs), associated with an altered morphology of the mitochondrial cristae and a dampened expression of different subunits of the mitochondrial respiratory chain. Interestingly, in both studies, the ATP/ADP ratio was not significantly altered by *PSEN2* ablation, likely because of a compensatory upregulation of the glycolytic flux [90]. Recently, we obtained data suggesting that, in primary cortical neurons from PS2^−/−^ mice (see Box 1), reduced mitochondrial respiration is associated with a defective mitochondrial Ca^2+^ signal (Rossi et al., in preparation). This finding suggests that the Ca^2+^-mediated modulation of mitochondrial metabolism has a key role in the effects reported above [91]. 

It is well established that mitochondrial activity is critical for brain health—not only is the majority of neuronal ATP synthesized by mitochondria, but also the rate of ATP synthesis matches synaptic activity [92]. Therefore, the effects of FAD-PS on mitochondria metabolism might be relevant to FAD pathogenesis. 

Alterations of mitochondrial activity have been reported in different AD models, mostly in Tg mouse models harboring FAD-PS1 and FAD-APP mutations. However, consensus has not been reached on the underlying molecular mechanisms. Indeed, either defective assembly/expression/activity of different subunits of the mitochondrial respiratory chain [93], altered mitochondrial Ca^2+^ signals [94,95], or organelle positioning/transport [96] have been suggested to contribute to the observed alterations. In contrast, only a few studies have focused on FAD-PS2 mutants. In primary cortical neurons from 2TG mice (see Box 1), we observed a reduced mitochondrial respiratory capacity [97]. This defect is not due to any intrinsic alteration of the respiratory chain, but rather depends on an impaired glycolytic flux, in turn affecting nutrient supply to mitochondria and thus organelle metabolism. However, considering that 2TG mice also express the FAD-APP mutant, it is not clear to what extent FAD-PS2 contributes to this phenotype. Recently, however, in different FAD-PS2-expressing cells, we observed a lower mitochondrial activity associated with a reduced ATP synthesis [98]. Mechanistically, these alterations depend in part on reduced mitochondrial Ca^2+^ signalling (due to partial depletion of ER Ca^2+^ content; see above), and in part on defective mitochondrial pyruvate uptake, caused by alterations in a signalling pathway driven by hyperactive GSK3β [98] (Figure 2), a feature commonly reported in AD [99]. Importantly, when compared to WT, in primary cortical neurons from TG mice (see Box 1), basal ATP levels are not significantly affected, whereas a faster ATP decrease is observed in cells exposed to ATP-consuming stimuli. In addition, we found that these metabolic alterations are associated with an increased susceptibility of FAD-PS2-N141I neurons to excitotoxicity induced by glutamate at physiological concentrations [98]. Overall, these results suggest that subtle mitochondrial alterations may be tolerated for a long time until specific stress conditions, imposing a high energy-demand, unveil their pathological potential. This might be relevant in neurological disorders characterized by a late onset, such as AD.

### 3.3. Brain Network Activity

Ca^2+^ dysregulation and altered APP processing, the two major hits linked to PS2-N141I expression, could affect neural circuit dynamics during the progression of amyloidosis. By studying brain oscillatory activity of adult 2TG mice under anesthesia, we observed that these mice develop a condition of hippocampal hyperactivity, with increased power in the gamma frequency range (45–90 Hz), as measured by spontaneous local field potential (LFP) signals. Curiously, age-matched TG mice also show a similar increase in the gamma power [100]. This hyperactivity is thus independent of Aβ production given that TG mice, unlike the 2TG animals, show neither plaque deposition nor gliosis, and Aβ42 levels are not significantly different from those found in WT mice [100]. This also suggests that, in 2TG mice, network hyperactivity is not due to compensatory, protective mechanisms and likely exerts a pathogenic role in the disease [78]. Of note, in humans, mild cognitive impairment (MCI) is marked by hyperactivity in the hippocampus, as well as in other cortical regions, that disappears with overt AD [101]. 2TG mice also present hyper-synchronicity, which is detectable as early as 3 months of age [100]. This aspect is likely attributable to the early phase of Aβ accumulation and represents a common feature in AD, often in the form of silent seizures, especially in FAD cases that show a higher incidence of epilepsy [102,103,104].

Studies on the brain electrical activity of both AD patients and mouse models have recently been focused on slow oscillations, which are directly involved in memory consolidation during sleep and unconsciousness [105]. By detecting mesoscale Ca^2+^ signals at the mouse brain level, Busche and coworkers elegantly demonstrated that functional connectivity in the slow-wave range (0.1–3 Hz) is severely reduced in the neocortex, thalamus, and hippocampus of different AD mouse models, also on the basis of PS1 [106]. Interestingly, slow-wave manipulation restores the functionality of brain circuits, rescues neuronal Ca^2+^ [107], and enhances memory consolidation in both types of mice [106,107].

Given that PS2-N141I alters neuronal and astrocytic Ca^2+^ homeostasis, it might also disturb hippocampal and cortical oscillatory activity in the slow-wave range. In mice under anesthesia, the oscillatory activity of different brain depths can be measured by simultaneously recording LFP signals with a multi-site linear probe. We used this approach to study brain rhythmicity at the cortical and hippocampal levels. In both TG and 2TG mice, the total power, which mostly reflects spontaneous activity in the low frequency range (0.1–5 Hz), is reduced, particularly at the hippocampal level, suggesting that the PS2 mutant by itself alters the brain electrical activity [108]. 

Another interesting feature shared by both TG and 2TG mice is the disruption of cortico-hippocampal oscillation coupling (Figure 2, [108]). The phenomenon, also known as phase-amplitude coupling (PAC), occurs when the phase of slower rhythms influences the amplitude of faster ones, and it has been found to be involved in memory consolidation and information transfer [105,109]. 

Unique features of 2TG mice help to mark the progression of Aβ accumulation and deposition—loss of functional connectivity in the slow-wave range marks the onset of Aβ accumulation, similarly to what reported in PS1-based AD mice [106], whereas low/high power imbalances characterize Aβ deposition in plaque-seeding mice [108]. Since Aβ42 oligomers are associated with Ca^2+^ homeostasis dysregulation [110,111,112,113], it is tempting to speculate that, in 2TG mice, Ca^2+^ handling alterations, due to PS2-N141I, sum up or synergize with defects linked to Aβ accumulation.

## 4. Concluding Remarks and Possible Therapeutic Targets

At the brain circuit level, the FAD-linked PS2-N141I mutant increases excitability [54,100] and disrupts the coupling of cortical slow-waves to hippocampal fast gamma frequencies [108]. Altogether, these findings are consistent with the high frequency of seizures and behavioral changes found in both FAD-PS2-N141I patients [1] and other mouse models expressing PS2-N141I [114,115]. From an pathogenic point of view, major alterations are expected in subpopulations of fast spiking interneurons that control the excitability of neuronal microcircuits, as reported in AD mouse models [103,116,117,118]. These highly active cells are likely more susceptible to the metabolic failure brought about by the aforementioned defective mitochondrial function [97,98]. One should also consider that, in these mouse models, only the PS2 mutant is expressed in both neurons and glial cells. In particular, astrocytes are good candidates to explain circuit dysfunctions given that, through spontaneous Ca^2+^ oscillations and intercellular Ca^2+^ waves, they can control the excitability of large neuronal networks [119,120], as well as modulate neighboring neurons by glio-transmission [121,122]. Furthermore, Ca^2+^ dysregulation and metabolic impairment in a cell type can also affect the closest cells, thus necessitating their investigation at the in situ and in vivo level. 

It can be speculated that defects in metabolic and autophagic pathways, directly dependent on Ca^2+^ dysregulation (see above), are responsible for the described network hyperexcitability and excitation/inhibition imbalances, which has also been reported in other AD models [77,103,123]. 

As for Ca^2+^ dysregulation, a common denominator between FAD-PS1 and -PS2 mutations is SOCE. Nevertheless, up until now, only a few studies have addressed the role of this Ca^2+^ pathway in neurons, mainly because of technical problems, i.e., the difficulty of distinguishing between activation of SOCE and voltage-operated Ca^2+^ channels (VOCCs) [124,125]. Recently, STIM2 and ORAI2, two key players in SOCE machinery, have emerged as key components of neuronal SOCE, being implicated in SOCE impairment in mushroom spines of hippocampal neurons from FAD-PS1-M146V knock-in mice [126,127,128]. Although the role of SOCE in neurons is still unclear, it is important to stress that, in excitable cells, STIM and ORAI components might also play non-canonical roles—STIM1 binds to L-type VOCCs, inhibits their gating, and induces channel internalization [129,130] while ORAI1 increases neuronal excitability [131,132]. At variance with neuronal cells, it is now largely accepted that SOCE is crucial for the Ca^2+^-based excitability that characterizes glial cells both in vitro [133,134] and in vivo [135]. Nonetheless, studies that specifically address the role of FAD-PS2 in glial SOCE modulation are still lacking. Considering also the complexity of microglia involvement in the onset and progression of AD [113,136], it is conceivable that these cells might also be primarily affected in the SOCE pathway, given that PS2 is the major core component of γ-secretase complexes expressed in this cell type [137].

We have recently shown that there is an inverse relationship between SOCE level and Aβ42 accumulation [138], consistent with data obtained in neurons [70] and other model cells [139]. These observations suggest the possibility of rescuing the SOCE defect in neural cells while antagonizing Aβ42 production. It has been demonstrated that, in mouse lymphocytes, SOCE is increased by knockout of ORAI2, a channel subunit and a negative modulator of SOCE that is responsible for the Ca^2+^ release-activated Ca^2+^ current [140]. In Aβ42-secreting neuroglioma cells, ORAI2 downregulation also increases SOCE and reduces the Aβ42/Aβ40 ratio [138] (Figure 2). Of note, astrocytes actively participate in Aβ production and clearance [141]. We do not know yet whether ORAI2 can play a similar role in neurons; up until now, it looks unlikely, given that recent data by Betzprozvanny’s group favor the hypothesis that ORAI2 is a component of a specific type of neuronal SOCE that is based on transient receptor potential canonical 6 (TRPC6) channel and regulated by diacylglycerol [127]. What is clear is that investigating Ca^2+^ dysregulation in AD allows the design of alternative therapeutic approaches to this devastating disease.

Additional therapeutic approaches could be suggested on the basis of altered bioenergetic and autophagy pathways. Impaired mitochondria, unable to supply cellular ATP demand, cause alterations in neuronal excitability, eventually leading to Ca^2+^ overload and cell death [142]. Moreover, the accumulation of damaged mitochondria (and misfolded proteins), due to defective autophagy, further contributes to dysfunctional neurons, causing, over the long term, neurodegeneration. Indeed, mitochondrial alterations, and in particular defects in bioenergetic pathways, have been widely reported to be key factors not only in AD but also in other neurodegenerative diseases [142,143]. Importantly, bioenergetic alterations are reported in different SAD and FAD samples, appearing at the early stage of the disease, before Aβ plaque formation [144].

The bioenergetic state of neurons is a crucial determinant of their response to glutamate, with cells containing defective mitochondria undergoing bioenergetic crises, Ca^2+^ mishandling, and excitotoxicity. The Food and Drug Administration (FDA)-approved molecule memantine targets glutamate receptors and is among the few pharmacological treatments that provide modest benefits in AD patients, in addition to cholinesterase inhibitors [145]. Targeting Ca^2+^ defects, at multiple levels, was suggested as a possible therapeutic strategy, especially in the form of drug repurposing. Among the best candidates, there are dantrolene, a RyR modulator [146], and isradipine, a VOCC inhibitor, as reviewed by Chakraborty and Stutzmann [147]. Attention has to be payed to the fact that dihydropyridines, especially nimodipine, also increase Aβ42 secretion [148]. None of these drugs are in the pipeline yet, and thus additional interventions aiming at supporting other pathways, such as mitochondrial performance, are desirable. In line with this, we showed that GSK3β inhibition rescues the FAD-PS2-linked bioenergetic defect [98]. Interestingly, both PS and Aβ oligomers have been reported to interact with the kinase, favoring its activity [99]. Considering the fact that GSK3β activity has been observed at MAM [149], where PSs are also enriched and Aβ peptides are generated [63,65,66], a MAM-targeted intervention might represent a useful therapeutic strategy [98].

Finally, the impairment in mitochondrial bioenergetics described in AD models is likely linked to a metabolic rewiring, possibly resulting in systemic alterations in the concentration of specific metabolites. Thus, the detailed metabolic profiling of AD patient-derived peripheral samples (blood and cerebrospinal fluid) might offer the possibility to discover new biomarkers that are helpful for early AD diagnosis, as has been previously suggested [150,151].

## 5. Box 1: AD Mouse Models Based on PS2

Several mouse models have been developed to understand the pathogenesis of AD, however, none of them are capable of reproducing the full spectrum of the human disease. The large majority of the most used AD models are double-Tg mice based on human FAD-APP and -PS1 mutations, both required to obtain fast amyloid accumulation, plaque deposition, and gliosis between 2 and 8 months of age. These Tg mice are widely considered to be adequate models of Aβ amyloidosis and its inflammatory process; they allow us to study the initial stages of the disease, according to the vision that places Aβ toxicity among the first hits in the AD cascade [6,152,153]. Nonetheless, the latter appears necessary but not sufficient in terms of causing neurodegeneration, with other concomitant and downstream factors playing a key role [7]. Neurodegeneration, linked to tau aggregation, is in fact mainly present in 3xTg-AD mice, which host three human mutant genes encoding PS1, APP, and tau [154].

Curiously, only the PS2-N141I mutation has been used to generate AD mouse models based on *PSEN2*. In terms of the latter, we used two homozygous lines: the double Tg (2TG) B6.152H, also known as B6.PS2APP, and the single Tg (TG) PS2.30H [76]. The latter line expresses the human PS2-N141I under the *prion protein* promoter, with background C57Bl/6 > 90% [155]. The B6.152H line was instead obtained by co-injection of human *PSEN2,* carrying the N141I mutation—under the mouse *prion protein* promoter—and the human *APP* isoform 751, carrying the APP-KM670/671NL Swedish mutation—under the *Thy1.2* promoter—into zygotes of the C57Bl/6 strain (background C57Bl/6 100%) [76].

The PS2.30H line was originally used to obtain hemizygous PS2APP mice by crossing PS2.30H females with APP-Swedish males of the BD.AD147.71H line, with background C57Bl/6 > 90% [155]. Up to 12 months of age, TG mice show neither plaques nor Aβ accumulation in the brain [100]. The histopathological traits of PS2APP and B6.PS2APP are very similar, showing an exponential growth of Aβ accumulation and plaques at 3 and 6 months of age, respectively [76,155]. Plaque deposition starts in the frontal cortex, subiculum, and hippocampus; increases for up to 12–16 months of age; and correlates with the level of human APP transcript [76,155]. Behavioral deficits have only been characterized thoroughly in PS2APP mice, with spatial learning (Morris water maze) and memory defects appearing at 8 months [155]. Biochemical and functional differences between the two closely related models are also present [156,157]. In our studies, TG and 2TG mice are maintained and used in homozygosity, a condition that allows for the reduction of the variability of APP expression [76]. The two lines express PS2 at a similar level, about twice that found in C57Bl/6 WT mice, used as controls [54].

B6.152H mice have also been used in hemizygosity (B6.152) to study different aspects of the AD phenotype [158], or to generate TauPS2APP triple Tg mice, upon crossing the B6.152H line with the Tau-overexpressing pR5 line [159,160] that expresses the human tau-40 isoform under the *Thy1.2* promoter [161]. Of note, a PS2-/- mouse line has been obtained by neomycin insertion in the C57Bl/6 × 129Sv genetic background [162,163]. This line does not show alterations of the endogenous APP processing and it is useful in terms of studying the physiological role of PS2 and possible loss-of-function defects associated with PS2-N141I expression [100,108]. It was also used to produce *PSEN* conditional double KO mice and study neurodegeneration and memory impairments due to PS deficiency [164].

Other AD mouse models, based on PS2-N141I, have been generated under different promoters and genetic backgrounds. Comparison of their histopathological, functional, and behavioral properties is beyond the scope of this review. The latest generation of AD mouse models avoids the overexpression of FAD mutations and is focused on risk genes, such as Triggering receptor expressed on myeloid cells 2 (*TREM2*) [165] and Apolipoprotein E4 (*APOE4*) [166]. More than 200 AD mouse models are now available; detailed information about these AD animal models is available at the Alzforum website (https://www.alzforum.org/). It is also necessary to mention that doubts have recently been raised on the use of Tg mice to study human AD, given that, at variance with the trascriptomic profiles of physiological human and rodent brain aging, which appear very similar, those of AD brains are largely different between humans and rodents, and even between different Tg AD mouse lines [167].

## Figures and Tables

**Figure 1 cells-09-02166-f001:**
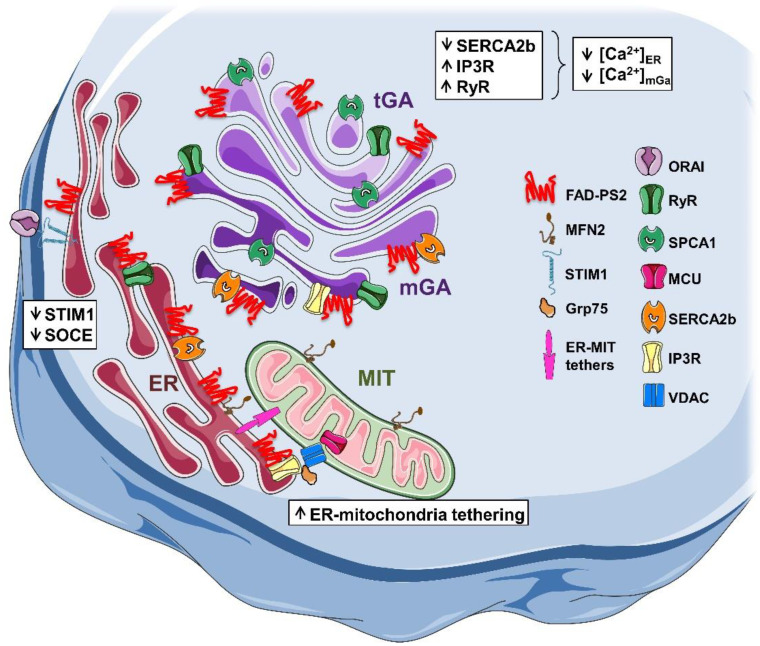
Familial Alzheimer’s disease (FAD)-presenilin-2 (PS2) alters multiple Ca^2+^ signalling pathways. The cartoon represents different intracellular membrane localizations of FAD-PS2, its interactions with several components of the molecular Ca^2+^ toolkit, and multiple Ca^2+^ signalling pathways that are altered by its action. See text for details. ER, endoplasmic reticulum; mGA, medial-Golgi apparatus; tGA, trans-Golgi apparatus; MIT, mitochondrion.

**Figure 2 cells-09-02166-f002:**
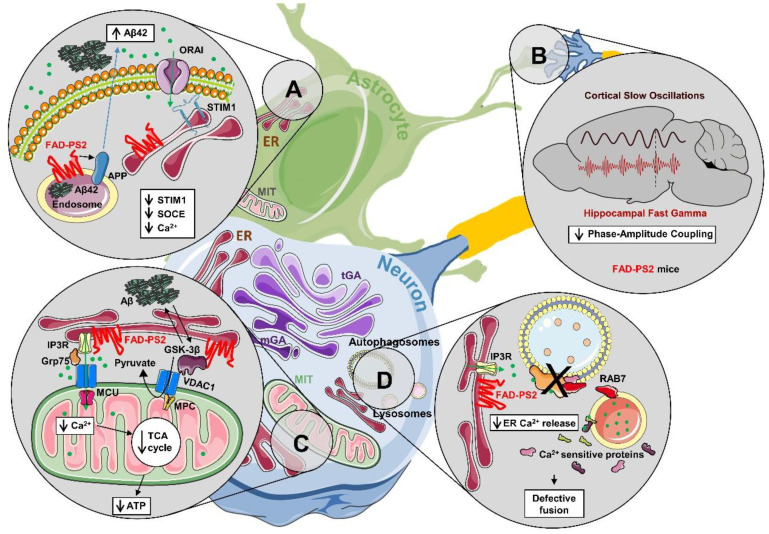
Functional consequences of dysregulated Ca^2+^ signalling induced by FAD-PS2. The cartoon represents the major dysfunctions linked to the expression of FAD-PS2 mutants at both the cellular and brain network levels. (**A**) Decreased store-operated Ca^2+^ entry (SOCE) potentiates amyloid precursor protein (APP) processing and Aβ42 production. (**B**) FAD-PS2-N141I-based mice show altered neuronal circuits (decreased phase-amplitude coupling between cortical slow oscillations and hippocampal fast gamma frequencies). (**C**) Decreased mitochondrial Ca^2+^ signalling and pyruvate uptake impair mitochondrial metabolism and cell bioenergetics. (**D**) Reduced endoplasmic reticulum (ER) Ca^2+^ release blocks the recruitment to autophagosomes of the Ras-associated binding protein RAB7 and their subsequent fusion with lysosomes. See text for details.

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
