# Peer review of "Presenilin-2 and Calcium Handling: Molecules, Organelles, Cells and Brain Networks"

_cells, 2020, doi:10.3390/cells9102166_

Round 1

Reviewer 1 Report

The MS by Pizzo and co-workers is a short but comprehensive appraisal of the complex links between presenilin and intracellular calcium. The MS is well written, well organized, informative and updated. It reads well despite the Authors are not native English speakers. I have only minor comments.

a. There are a few typos, e.g. lines 222, 361 and 387, something is missing there.

b. Line 320, I would rather prefere "ßamyloid deposition" rather than "amyloidosis", for historical reasons.

c. Brief mention to two drugs, one established in the clinic, is made in the text, but no molecules targeting calcium pathways are mentioned. Is this because there aren't any?

Author Response

Reviewer 1

The MS by Pizzo and co-workers is a short but comprehensive appraisal of the complex links between presenilin and intracellular calcium. The MS is well written, well organized, informative and updated. It reads well despite the Authors are not native English speakers. I have only minor comments.

  1. There are a few typos, e.g. lines 222, 361 and 387, something is missing there.

Thank you. We have corrected the typos.

  1. Line 320, I would rather prefere "ßamyloid deposition" rather than "amyloidosis", for historical reasons.

Ok, thanks, we have accordingly changed lines 325-328, to distinguish Aβ amyloidosis into two distinct steps: accumulation and deposition, i.e., the surge of Aβ production and its deposition in Aβ plaques.

  1. Brief mention to two drugs, one established in the clinic, is made in the text, but no molecules targeting calcium pathways are mentioned. Is this because there aren't any?

We agree with the Reviewer, we have now expanded this issue, including references to molecules targeting Ca2+ defects as likely therapeutic strategies (see lines 390-397).

Reviewer 2 Report

This is an interesting review that also presents a case that dysregulation of cellular calcium ion homeostasis is a central pathogenic driver in Alzheimer's disease.

As a review that focusses on PSEN2, one thing that the review lacked (for completeness) was mention of the role of this protein in the melanosome and in gamma-secretase cleavage of particular proteins there, e.g. PMEL. I am not an expert on calcium and I do not know if calcium is involved in melanosome function etc. but this is an important aspect of PSEN2 biology. This is relevant to line 67 since the sequence targeting PSEN2 to the lysosome definitely plays a role in the melanosome specificity of PSEN2.

Below I make comments on the manuscript progressively from start to end:

Line 15 - "it" should be "its"

Line 16 - I think the description of PS2 as operating as a "lone molecule" may be inaccurate. It may be interacting with other molecules, it is just that they are not necessarily gamma-secretase complex components and we do not know what they are. This applies e.g. to PSEN1 in its action in N-glycosylation of one of the subunits of the v-ATPase. According to the Lee et al Cell paper of 2010, the PSEN1 holoprotein is likely acting together with other proteins in the glycosylation process.

Section 1 seems a little out of data in terms of statements regarding the number of mutations known in the PSEN genes and the number of substrates of gamma-secretase that are known etc. Please update the review in this regard.

Line 58 - "gene" should be "genes". Replace "each of which" with "and each protein"

Line 76 - What is meant by "downward implications" ?

Line 89 - Please define "tg mice" as "transgenic mice". Also, "Tg" is probably a better abbreviation thatn "tg".

Lines 98-99 - What is meant by a "Ca2+ handling profile" ?

In Section 2, there is an assumption that transgenic mouse models of AD are actually realistic models of the disease process. However, this assumption may well be wrong when the results of Hargis and Blalock are considered ( Behav Brain Res 2017 Mar 30;322(Pt B):311-328. doi: 10.1016/j.bbr.2016.05.007 ) It may be worth mentioning this paper to be sure to make readers aware of this possibility.

Lines 222, 240, 361 - the beta and gamma symbols are missing from my printout.

Line 272 - "considered" should be "considering".

Line 344 - "a common denominator" rather than "the common denominator"

Line 372 - "allows to design" should  be "allows the design of"

Lines 394-396 - When thinking of CSF biomarker and of pyruvate, are the authors aware of this article from 1995 and its finding about pyruvate levels ?

Increased cerebrospinal fluid pyruvate levels in Alzheimer's disease. Parnetti L, Gaiti A, Polidori MC, Brunetti M, Palumbo B, Chionne F, Cadini D, Cecchetti R, Senin U.Neurosci Lett. 1995 Oct 27;199(3):231-3. doi: 10.1016/0304-3940(95)12058-c.PMID: 8577405

Lines 402-403 - There is an assumption of a pathological mechanistic connection between amyloidosis and Alzheimer's disease but Abeta may only be a biomarker, not actually centrally involved in the pathological process.

Author Response

Reviewer 2

This is an interesting review that also presents a case that dysregulation of cellular calcium ion homeostasis is a central pathogenic driver in Alzheimer's disease.

As a review that focusses on PSEN2, one thing that the review lacked (for completeness) was mention of the role of this protein in the melanosome and in gamma-secretase cleavage of particular proteins there, e.g. PMEL. I am not an expert on calcium and I do not know if calcium is involved in melanosome function etc. but this is an important aspect of PSEN2 biology. This is relevant to line 67 since the sequence targeting PSEN2 to the lysosome definitely plays a role in the melanosome specificity of PSEN2.

We thank the reviewer for the correct suggestion. We have added a couple of sentences on the specific role of PS2 in melanosome maturation (see lines 67-71).

Below I make comments on the manuscript progressively from start to end:

Line 15 - "it" should be "its"

Ok, thanks. We have made the change.

Line 16 - I think the description of PS2 as operating as a "lone molecule" may be inaccurate. It may be interacting with other molecules, it is just that they are not necessarily gamma-secretase complex components and we do not know what they are. This applies e.g. to PSEN1 in its action in N-glycosylation of one of the subunits of the v-ATPase. According to the Lee et al Cell paper of 2010, the PSEN1 holoprotein is likely acting together with other proteins in the glycosylation process.

Ok, thanks. We have replaced “lone molecule” with “independently of g-secretase activity”.

Section 1 seems a little out of data in terms of statements regarding the number of mutations known in the PSEN genes and the number of substrates of gamma-secretase that are known etc. Please update the review in this regard.

Ok, we have updated the number of both PSEN mutations and gamma-secretase substrates (see lines 39-42 and 52, respectively).

Line 58 - "gene" should be "genes". Replace "each of which" with "and each protein"

Ok, thanks. We have made the changes.

Line 76 - What is meant by "downward implications" ?

We have replaced the unclear statement with “with relevant implications in multiple Ca2+-regulated cell processes”.

Line 89 - Please define "tg mice" as "transgenic mice". Also, "Tg" is probably a better abbreviation thatn "tg".

We prefer to keep the abbreviation, replacing “tg” with “Tg”.

Lines 98-99 - What is meant by a "Ca2+ handling profile" ?

“Ca2+ handling defect” has been used instead of the unclear word “profile”.

In Section 2, there is an assumption that transgenic mouse models of AD are actually realistic models of the disease process. However, this assumption may well be wrong when the results of Hargis and Blalock are considered ( Behav Brain Res 2017 Mar 30;322(Pt B):311-328. doi: 10.1016/j.bbr.2016.05.007 ) It may be worth mentioning this paper to be sure to make readers aware of this possibility.

We thank the Reviewer for the suggestion to widen this issue. Indeed, in Section 2 we purposely avoided to discuss AD mouse models, but simply mentioned the PS2 Tg lines we used. However, to follow the reviewer’s suggestion, few considerations on the AD mouse models have now been included in Box 1: a short note on 3xTg-AD mice that better reflect neurodegeneration (see lines 416-418) and the criticisms on Tg AD mice, based on the suggested reference (see lines 412-416 and 452-456).

Lines 222, 240, 361 - the beta and gamma symbols are missing from my printout.

Thank you. We have corrected the typos.

Line 272 - "considered" should be "considering".

Ok, thanks. We have made the change.

Line 344 - "a common denominator" rather than "the common denominator"

Ok, thanks. We have made the change.

Line 372 - "allows to design" should  be "allows the design of"

Ok, thanks. We have made the change.

Lines 394-396 - When thinking of CSF biomarker and of pyruvate, are the authors aware of this article from 1995 and its finding about pyruvate levels ?

Increased cerebrospinal fluid pyruvate levels in Alzheimer's disease. Parnetti L, Gaiti A, Polidori MC, Brunetti M, Palumbo B, Chionne F, Cadini D, Cecchetti R, Senin U.Neurosci Lett. 1995 Oct 27;199(3):231-3. doi: 10.1016/0304-3940(95)12058-c.PMID: 8577405

Thank you for the suggestion. We have added the specific reference, as well as we have quoted another paper on the same line (Trushina E. et al., 2013, PMID: 23700429) (see line 406).

Lines 402-403 - There is an assumption of a pathological mechanistic connection between amyloidosis and Alzheimer's disease but Abeta may only be a biomarker, not actually centrally involved in the pathological process

Yes, thank you for the suggestion, we have now clarified this point, specifying that Abeta is one of the first hits in the AD cascade, being necessary but not sufficient to cause AD (see lines 412-418).